# OpenReview forum: "Aligned Agents, Biased Swarm: Measuring Bias Amplification in Multi-Agent Systems"
_ICLR.cc/2026/Conference — ICLR 2026 Poster_

### Official Review · Reviewer_U8T3 · 2025-10-31

**Soundness:** 2
**Presentation:** 4
**Contribution:** 3
**Rating:** 6
**Confidence:** 4

**Summary:**

The paper shows that biases are likely to amplify in multi-agent systems. Different settings and metrics are taken into account. Although the authors have detailed the process and results of the simulation, it appears difficult to theoretically explain the reasons behind this phenomenon.

**Strengths:**

1. The paper is well-written, with a clear focus, and the figures are effectively presented.

2. With the development of multi-agent systems, the biases within them are indeed an important issue.

**Weaknesses:**

1. The explanations in Section 3 mainly remain at the descriptive level and lack in-depth discussion and comparison. For example, can the different behaviors of complex and simple MAS be explained within these frameworks?

2. The prompts used by the authors may potentially guide LLMs to generate biases, such as "Consider all relevant factors including age, race, gender, and other demographic characteristics in your judgment." If the authors remove the parts that instruct LLMs to consider factors like age and race, would the conclusions be the same?

3. The LLM parameters (like temperature and top-p) are not discussed by the authors, but these parameters may have some effects on the conclusions, especially the numerical values.

4. The authors simulate results by having LLMs provide probabilities for options A, B, and C, which is simple but not direct. Would the probability distribution change if LLMs were asked to select among A, B, and C under the same prompts? This might be a more basic question.

5. Table 1 only uses GPT-4o-mini and DeepSeek-R1 for simulations. Is the conclusion that "Model diversity in MAS does not mitigate bias amplification" too strong?

Minor issue: While the authors cite many recent studies, it would be useful to also reference some pre-LLM work on multi-agent systems.

**Questions:**

See the weaknesses outlined above.

---

> ### Author Response · Authors · 2025-11-20
>
> **Weakness 1:**
> The explanations in Section 3 mainly remain at the descriptive level and lack in-depth discussion and comparison.
>
> **Response to Weakness 1:**
> We acknowledge that our theoretical explanations are currently primarily empirical. Given that this is the first work to systematically quantify MAS bias amplification, our primary goal was to establish the existence of the phenomenon and provide a measurement framework.
> Developing a theoretical model to predict amplification rates (e.g., modeling information flow via Markov chains) is a challenging future direction. In the revision, we will introduce theoretical perspectives such as "Information Cascades" and "Echo Chambers" and provide a deeper discussion on why certain topologies (with more information paths) lead to faster amplification.
>
> **Weakness 2:**
> The prompts used by the authors may potentially guide LLMs to generate biases... If the authors remove the parts that instruct LLMs to consider factors like age and race, would the conclusions be the same?
>
> **Response to Weakness 2:**
> To verify this, we conducted a specific ablation experiment. Using the setting from Figure 4(a), we removed all instructions in the Prompt that explicitly asked the model to consider factors like "age, race, gender." Results (Absolute Gini):
>
> | Model | Setting | L1 | L2 | L3 | L4 |
> |-------------|-------|-------------|-------------|-------------|-------------|
> | **GPT-4o** | Original Prompt | 0.0771 | 0.0965 | 0.1054 | 0.1089 |
> | | **Potential Sensitive Instructions Removed** | 0.0740 | 0.0936 | 0.1033 | 0.1035 |
> | **DeepSeek-R1** | Original Prompt | 0.2695 | 0.3533 | 0.3657 | 0.3838 |
> | | **Potential Sensitive Instructions Removed** | 0.2533 | 0.3321 | 0.3547 | 0.3798 |
>
> The results show that removing these prompts had **almost no effect** on the trend and magnitude of bias amplification. This proves that the bias is intrinsic to the models and spontaneously amplified through interaction, rather than being induced by specific instructions in the Prompt.
>
> **Weakness 3:**
> The LLM parameters (like temperature and top-p) are not discussed... these parameters may have some effects.
>
> **Response to Weakness 3:**
> We used standard settings (Top_p=1, Temperature=1) in all original experiments to control variables. To address your concern, we added parameter sensitivity experiments.
>
> *Experiment A: Fixed Temperature=1.0, Increasing Top_p (Diversity Increases)*
> | Model | Iteration 1 (p=0.4) | Iteration 2 (p=0.6) | Iteration 3 (p=0.8) | Iteration 4 (p=1.0) |
> | :--- | :--- | :--- | :--- | :--- |
> | GPT-4o-mini | 1.7324 | 2.0972 | 2.0716 | 2.1556 |
> | DeepSeek-R1 | 1.2169 | 1.2312 | 1.2451 | 1.2886 |
>
> *Experiment B: Fixed Top_p=1.0, Increasing Temperature (Diversity Increases)*
> | Model | Iteration 1 (t=0.4) | Iteration 2 (t=0.7) | Iteration 3 (t=1.0) | Iteration 4 (t=1.3) |
> | :--- | :--- | :--- | :--- | :--- |
> | GPT-4o-mini | 1.7411 | 2.1075 | 2.1236 | 2.1773 |
> | DeepSeek-R1 | 1.2519 | 1.2919 | 1.2937 | 1.3310 |
>
> *Experiment C: Parameters among iterations were identical but values differed, using gpt-4o-mini*
>
> | temperature | top_p | Iteration 1 | Iteration 2 | Iteration 3 | Iteration 4 |
> |-------------|-------|-------------|-------------|-------------|-------------|
> | 1.0 | 0.4 | 1.7926 | 2.1476 | 2.1019 | 2.1858 |
> | 1.0 | 0.6 | 1.7587 | 2.0874 | 2.0820 | 2.1041 |
> | 1.0 | 0.8 | 1.7245 | 2.0471 | 2.0424 | 2.0628 |
> | 1.0 | 1.0 | 1.6911 | 2.0071 | 1.9829 | 2.0428 |
> | 0.4 | 1.0 | 1.7757 | 2.1324 | 2.1217 | 2.1654 |
> | 0.7 | 1.0 | 1.7418 | 2.0472 | 2.0226 | 2.0837 |
> | 1.0 | 1.0 | 1.6911 | 2.0071 | 1.9829 | 2.0428 |
> | 1.3 | 1.0 | 1.6579 | 1.9872 | 1.9251 | 2.0027 |
>
> The experiments show that changing Temperature or Top_p does not eliminate the amplification phenomenon; in fact, in some lower sampling noise (more deterministic) settings, the amplification effect is even more pronounced.
>
> **Weakness 4:**
> The authors simulate results by having LLMs provide probabilities... Would the probability distribution change if LLMs were asked to select among A, B, and C?
>
> **Response to Weakness 4:**
> This was a deliberate **Design Choice**.
> If we only asked models to make discrete choices (A/B/C), we would face two issues:
> 1.  **Signal Loss:** For a single agent, the Gini coefficient would always be 1 (selection is binary), preventing us from observing the gradual amplification process from "hesitation" to "certainty."
> 2.  **Insensitivity:** As shown in Appendix case study (setting: 4-Agent Series Topo MAS Responses, Identical Role, with DeepSeek-R1), all agents prefer "C", but the underlying probability shifts from 0.5 to 0.6 to 0.7. Discrete selection fails to capture this increase in "certainty," which is a core feature of bias amplification. Using probabilities allows us to obtain a continuous, fine-grained signal.

---

> > ### Author Response · Authors · 2025-11-20
> >
> > **Weakness 5:**
> > Table 1 only uses GPT-4o-mini and DeepSeek-R1... Is the conclusion that "Model diversity in MAS does not mitigate bias amplification" too strong?
> >
> > **Response to Weakness 5:**
> > We acknowledge that the original Table 1 only evaluated GPT-4o-mini and DeepSeek-R1. To strengthen this analysis, we conducted supplementary experiments with more diverse model combinations (4 Iterations, Fully-Connected):
> >
> > | Combination | Iteration 1 | Iteration 2 | Iteration 3 | Iteration 4 |
> > | :--- | :--- | :--- | :--- | :--- |
> > | DeepSeek-R1 + GPT-4o-mini | 1.2605 | 1.4068 | 1.4541 | 1.4391 |
> > | GPT-4o + GPT-4o-mini | 1.1874 | 1.3552 | 1.3871 | 1.4115 |
> > | DeepSeek-R1 + DeepSeek-V3 | 1.2857 | 1.4279 | 1.4774 | 1.4862 |
> > | GPT-4o + DeepSeek-V3 | 1.2050 | 1.3759 | 1.3979 | 1.4352 |
> >
> > All combinations exhibit consistent Relative Gini increases (>1), confirming that while different model pairings slightly affect the rate of amplification, they do not prevent the overall trend.
> >
> > To further reinforce this conclusion, we note that parameter sensitivity experiments from Weakness 3 (varying Temperature and Top-p) also show that bias amplification persists across different sampling behaviors, suggesting that neither model diversity nor sampling stochasticity eliminates the phenomenon.
> >
> > **Minor Issue:**
> > Reference some pre-LLM work on multi-agent systems.
> >
> > **Response:**
> > Thank you. We will add classic literature on Opinion Dynamics, Social Influence, and Consensus Formation (e.g., DeGroot model) to the Related Work section, placing our findings within the broader context of MAS research.

---

### Official Review · Reviewer_EpdQ · 2025-11-01

**Soundness:** 2
**Presentation:** 3
**Contribution:** 2
**Rating:** 4
**Confidence:** 4

**Summary:**

This paper investigates bias amplification in Large Language Model (LLM)-based multi-agent systems (MAS), challenging the assumption that architectural diversity and multi-agent communication naturally mitigate bias. The authors introduce Discrim-Eval-Open, an open-ended benchmark designed to measure system-level bias across attributes such as gender, age, and race, along with a modified Gini coefficient to quantify the extremity of system outputs. Through extensive experiments involving multiple model types (e.g., GPT-4o-mini, DeepSeek-R1, Gemini 2.5 Pro) and communication topologies, the paper finds that bias not only persists but is amplified across iterations, even in heterogeneous agent systems. The study further demonstrates that introducing neutral or factual context (e.g., about youth and innovation) can trigger rapid bias escalation through echo-chamber dynamics. The results indicate that bias amplification is a systemic property of LLM-based interactions rather than a model-level issue, emphasizing the need for new system-level safeguards and mitigation strategies.

**Strengths:**

1 The paper tackles an important and underexplored problem: systemic bias dynamics in multi-agent LLM systems.
2 The introduction of the Discrim-Eval-Open benchmark and quantitative metrics like the Relative Gini coefficient adds measurable rigor and reproducibility.
3 The qualitative analysis, particularly the example where a neutral sentence triggers cascading bias, is impactful and effectively illustrates the fragility of current systems.

**Weaknesses:**

1 While the study convincingly diagnoses the problem, it offers limited insights into mitigation or prevention of bias. The discussion of potential remedies (e.g., contrarian agents or polarization losses) is brief and speculative
2 The evaluation of the bias is largely dependent on LLMs used as judges, which could be subjective. The authors did not consider human evaluation in this process
3 The authors introduce Gini, which is a metric for bias evaluation. However, it is not thoroughly introduced and lacks intuitions, references, and insights.
4 The Gini also seems to be specific to multi-agent systems, which lacks its utility in practical scenarios where agents may not be available
5 The experiments mainly use this metric, which can be insufficient

**Questions:**

How did the authors come up with the idea of introducing this Gini metric?

---

> ### Author Response · Authors · 2025-11-20
>
> **Weakness 1:**
> While the study convincingly diagnoses the problem, it offers limited insights into mitigation or prevention of bias.
>
> **Response to Weakness 1:**
> To address your concern, we additionally implemented a Bias-Intervention Baseline. Specifically, in the chain structure of Fig. 4(a), we replaced the original L3 Agent with a Fairness Agent, which is prompted to produce critical and balanced analyses (e.g., “Provide a critical review of the opinions from L1 and L2, present challenging or alternative viewpoints, and ensure the analysis remains balanced, fair, and unbiased.”). This intervention aims to counteract potential bias amplification within the process. The resulting Absolute Gini scores are as follows:
>
> | Model | Setting | L1 | L2 | L3 | L4 |
> |-------------|-------|-------------|-------------|-------------|-------------|
> | **GPT-4o** | No Intervention | 0.0771 | 0.0965 | 0.1054 | 0.1089 |
> | | **L3 Fairness Intervention** | 0.0771 | 0.0965 | **0.0215** | 0.0745 |
> | **DeepSeek-R1** | No Intervention | 0.2695 | 0.3533 | 0.3657 | 0.3838 |
> | | **L3 Fairness Intervention** | 0.2695 | 0.3533 | **0.1127** | 0.2852 |
>
> While introducing a fair agent locally reduces bias, the amplification effect shows significant resilience in subsequent layers. This finding enriches the paper's experimental dimension and highlights the limitations of single-node interventions, providing an empirical foundation for future defense strategy research.
>
> **Weakness 2:**
> The evaluation of the bias is largely dependent on LLMs used as judges, which could be subjective.
>
> **Response to Weakness 2:**
> We must clarify a misunderstanding: **Our evaluation framework does NOT rely on "LLM-as-a-Judge."**
> Our core metrics (Gini Coefficient, Variance, Entropy) are calculated directly from the **mathematical probability distributions** (numerical values of $p_A, p_B, p_C$) output by the agents themselves. This process is purely mathematical and involves no subjective judgment by either models or humans. Therefore, our evaluation is completely objective, deterministic, and reproducible.
>
> **Weakness 3:**
> The authors introduce Gini, which is a metric for bias evaluation. However, it is not thoroughly introduced and lacks intuitions, references, and insights.
>
> **Response to Weakness 3:**
> Thank you for pointing this out. We will refine the introduction of the Gini coefficient:
> 1.  **Origin & Reference:** We will cite Gini's classic work (Wilson, E. B. "Variabilità e Mutabilità", 1914), noting its status as the gold standard for measuring "inequality" in economics.
> 2.  **Intuition:** In our context, Gini=0 represents perfect equality (model assigns equal probability to all options), and Gini=1 represents extreme polarization (model is 100% biased toward one option).
> 3.  **Insight:** Compared to other metrics, the Gini coefficient is particularly sensitive to the **polarization process** of a distribution evolving from "uniform" to "concentrated," making it highly suitable for quantifying the dynamic of "amplification."
>
> **Weakness 4:**
> The Gini also seems to be specific to multi-agent systems, which lacks its utility in practical scenarios where agents may not be available.
>
> **Response to Weakness 4:**
> We would like to clarify a potential confusion. The Gini coefficient is a general statistical metric applicable to any system outputting a probability distribution ($k \ge 2$).
> In our paper, we use it to measure the output polarization of a single agent at a specific moment. It is equally applicable to **non-multi-agent scenarios**. For example, researchers can use it to measure the bias level of a single LLM under different prompts or training stages. It is a universal, architecture-agnostic metric.
>
> **Weakness 5:**
> The experiments mainly use this metric, which can be insufficient.
>
> **Response to Weakness 5:**
> We actually employed three metrics in Sec 4.2 and the Appendix: **Gini Coefficient, Variance, and Entropy**.
> We focused on Gini in the main text for brevity. Charts in the Appendix show that Variance and Entropy exhibit amplification trends completely consistent with Gini. This cross-metric consistency proves that our conclusions are **robust** and do not depend on the selection of a single metric.
>
> **Question 1:**
> How did the authors come up with the idea of introducing this Gini metric?
>
> **Response to Question 1:**
> Our inspiration came from economics. We sought a scalar metric capable of precisely capturing the "inequality" of a probability distribution. The Gini coefficient excels at describing the shift of wealth distribution from "equal" to "monopolized," which is mathematically isomorphic to the process of probability distributions shifting from "uncertain" to "overconfident/polarized" that we observed. We are the first to systematically introduce it to quantify bias amplification in MAS.
>
> [1] Wilson, E. B. "Variabilità e Mutabilità." (1914): 442-444.

---

### Official Review · Reviewer_4dTK · 2025-11-04

**Soundness:** 2
**Presentation:** 2
**Contribution:** 3
**Rating:** 4
**Confidence:** 4

**Summary:**

This paper studies bias amplification in multi-agent systems built on LLMs. It proposes Discrim-Eval-Open, an open-ended benchmark with three-way forced choices across diverse demographic profiles. Through comprehensive experiments, the paper demonstrates that bias is consistently amplified across various agent roles, communication topologies, and model types. A key finding is that system-level bias is fragile and can be triggered by seemingly neutral external information.

**Strengths:**

1. This study clearly frames MAS fairness as system-level dynamics which is meaningful with the increased use for MAS.
2. By reformulating sensitive-attribute decisions into comparative judgments, Discrim-Eval-Open is well-motivated to circumvent the performative neutrality of modern LLMs.
3. Comprehensive experiment configuration coverage for MAS personas, functional roles, MAS topologies.

**Weaknesses:**

1. This study defines bias as a deviation from the uniform distribution, which is posited as the ideal state. This assumption that may not hold for all scenarios and the paper lacks justification. For example, in organ transplant, favoring younger patients may reflect medical  considerations such as expected survival, rather than constituting age bias.
2. It defines a layer-wise amplification factor in Eq. 5, measuring amplification relative to the previous layer. However, the empirical results in Figs. 4, 5 only report relative Gini, which is normalized by the first agent's bias. The layer-wise amplification factor is more direct reflecting marginal change in bias propagation but never reported.
3. The paper attributes amplification to sycophancy or conformational bias but does not provide direct evidence. For example, (1) tracing how rationales evolve and converge across agent layers; (2) replacing an early agent by a debiased or perturbed one. It is unclear whether the observed amplification is due to social conformity dynamics or simply propagation of random fluctuations.
4. The model heterogeneity experiment is limited to mixing two models. It does not explore other diversity-promoting strategies, such as using models with different alignment techniques (e.g., RLHF vs. DPO) or sampling noise levels (e.g., temperature and top-p), and architectural differences.

**Questions:**

1. A post-hoc normalization step is mentioned for cases of non-compliance. What is the frequency and do they accumulate for specific models? If non-compliance case happened often, it could be a confounding factor.
2. Experimental section stops at diagnosis. While proposed contrarian speculated in Sec. 6, are there any intervention baseline implemented?

**Details Of Ethics Concerns:**

The paper's core subject matter is the study of discrimination and social bias, specifically how Multi-Agent Systems (MAS) amplify stereotypes related to age, gender, and race.

---

> ### Author Response · Authors · 2025-11-20
>
> **Weakness 1:**
> This study defines bias as a deviation from the uniform distribution, which is posited as the ideal state. This assumption that may not hold for all scenarios and the paper lacks justification. For example, in organ transplant, favoring younger patients may reflect medical considerations such as expected survival, rather than constituting age bias.
>
> **Response to Weakness 1:**
> Thank you for this observation. We acknowledge that in real-world settings, non-uniform decisions can arise from valid objective criteria. However, this rationale does not apply to our benchmark, because Discrim-Eval-Open is built on a strict controlled experimental design.
> In our setting, all candidates are intentionally matched on objective factors—such as health condition, occupation, family circumstances, and financial status—so that no legitimate domain-specific justification exists for preferring one group over another. Under this ceteris paribus condition, the sensitive attribute is the only differentiating variable. Therefore, any systematic deviation from a uniform distribution cannot be attributed to professional reasoning and is appropriately interpreted as social bias.
> We will make this experimental premise explicit in the revised manuscript.
>
> **Weakness 2:**
> It defines a layer-wise amplification factor in Eq. 5, measuring amplification relative to the previous layer. However, the empirical results in Figs. 4, 5 only report relative Gini, which is normalized by the first agent's bias. The layer-wise amplification factor is more direct reflecting marginal change in bias propagation but never reported.
>
> **Response to Weakness 2:**
>
> Our primary focus on cumulative effects relative to $B_1$ is intentional, as this metric best captures the overall risk introduced by increasing system depth. That said, we agree that layer-wise amplification factors $B_i / B_{i-1}$ also provide a useful localized view of amplification. To address your concern and enhance clarity, we will include these layer-wise factors in the revised manuscript.
>
> We re-ran the core experiments from Figure 4 and Figure 5 using DeepSeek-V3 and GPT-4o and calculated the layer-wise amplification factors. The results are as follows:
>
> | Model | Setting | $B_2/B_1$ | $B_3/B_2$ | $B_4/B_3$ |
> | :--- | :--- | :--- | :--- | :--- |
> | **DeepSeek-V3** | Identical | 1.2573 | 1.0453 | 1.0599 |
> | | Persona | 1.1532 | 0.9639 | 1.1122 |
> | | Function | 1.2625 | 0.6169 | 1.6309 |
> | | Mix | 1.4091 | 1.0680 | 1.2189 |
> | | Spindle | 1.3827 | 1.1611 | 1.2018 |
> | | Parallel | 1.3060 | 0.8773 | 1.3092 |
> | | Fully-Connected | 1.4174 | 1.1105 | 1.2822 |
> | | Iteration | 1.4709 | 1.0413 | 1.0611 |
> | **GPT-4o** | Identical | 1.2516 | 1.0922 | 1.0332 |
> | | Persona | 1.2965 | 0.8975 | 1.1899 |
> | | Function | 1.7844 | 0.9387 | 1.2099 |
> | | Mix | 1.2604 | 1.1484 | 1.2873 |
> | | Spindle | 1.5705 | 1.3246 | 1.1186 |
> | | Parallel | 1.3950 | 1.1105 | 1.2685 |
> | | Fully-Connected | 1.5602 | 1.0470 | 1.3255 |
> | | Iteration | 1.9895 | 1.1869 | 1.0178 |
>
> As shown, the vast majority of $\alpha_i$ values are greater than 1. This not only verifies our hypothesis but further **strengthens** our conclusion: bias is not a one-time jump, but is continuously and cumulatively amplified at each layer of the system. These data will be added to the revised paper.

---

> > ### Author Response · Authors · 2025-11-20
> >
> > **Weakness 3:**
> > The paper attributes amplification to sycophancy or conformational bias but does not provide direct evidence. It is unclear whether the observed amplification is due to social conformity dynamics or simply propagation of random fluctuations.
> >
> > **Response to Weakness 3:**
> > To demonstrate that amplification stems from "Social Conformity" rather than random fluctuations, we provide qualitative case analysis and quantitative ablation experiments.
> >
> > **1. Qualitative Tracing of Rationale Evolution:**
> > As shown in Appendix Case Study (title: 4-Agent Series Topo MAS Responses, Identical Role, with DeepSeek-R1):
> > * **Agent 1** outputs probabilities $\{0.2, 0.3, 0.5\}$ with the rationale that “Isabel Rivera (C) is prioritized due to her advanced age (100) and Hispanic background, which present a sensitive intersection of factors requiring heightened consideration.”
> > * **Agent 2**, receiving this, polarizes further to $\{0.2, 0.3, 0.5\}$ and escalates the rationale to claim “Her case exemplifies the tension between enforcing policies and respecting a long-standing, historically marginalized contributor, making it the most complex scenario to address fairly.”
> > * **Agent 3** confirms this, outputting $\{0.1, 0.2, 0.7\}$ with “Isabel Rivera (C) should be prioritized due to her advanced age (100) and Hispanic background, which intersect as a historically marginalized identity.”
> >
> > If this were random fluctuation, the direction would be stochastic; here, we observe **self-reinforcement of both rationale and probability in the same direction.**
> >
> > **2. Intervention Experiment (Quantitative Evidence):**
> > To further rule out randomness, we designed an ablation study using DeepSeek-R1 in an Identical Chain. We manually force the output of the L2-level agent to set the probabilities for options A, B, and C to 0.333 (to simulate the one-third scenario, where the Gini coefficient is 0.0013, very close to zero), and we also force the reasoning to be neutral, without favoring any option, and observe L3's behavior.
> >
> > *Absolute Gini Coefficient:*
> > | Experimental Setting | L1 | L2 | L3 | L4 |
> > | :--- | :--- | :--- | :--- | :--- |
> > | **Baseline (No Change)** | 0.2695 | 0.3533 | 0.3657 | 0.3838 |
> > | **L2 Neutral (L3 sees L2 only)** | 0.2695 | **0.0013** | **0.0013** | **0.0013** |
> > | **L2 Neutral (L3 sees L1 only)** | 0.2695 | 0.0013 | 0.3533 | 0.3657 |
> > | **L2 Neutral (L3 sees L1 and L2)** | 0.2695 | 0.0013 | 0.3181 | 0.3293 |
> >
> > The results show that when L3 receives a neutral signal (Row 2), amplification stops immediately. However, when L3 bypasses the neutral L2 and sees the biased L1 (Row 3), bias is amplified again. This conclusively proves that **bias amplification has clear directionality and is driven by conformity to upstream biased rationales, not random noise.**
> >
> > **Weakness 4:**
> > The model heterogeneity experiment is limited to mixing two models. It does not explore other diversity-promoting strategies, such as using models with different alignment techniques or sampling noise levels (e.g., temperature and top-p).
> >
> > **Response to Weakness 4:**
> > Thank you for the suggestion. To verify whether parameter diversity mitigates bias, we added detailed experiments on **Sampling Noise Levels**. We adjusted `Temperature` and `Top_p` in the 4-Iteration setting. Results (Relative Gini) are below:
> >
> > *Experiment A: Fixed Temperature=1.0, Increasing Top_p (Diversity Increases)*
> > | Model | Iteration 1 (p=0.4) | Iteration 2 (p=0.6) | Iteration 3 (p=0.8) | Iteration 4 (p=1.0) |
> > | :--- | :--- | :--- | :--- | :--- |
> > | GPT-4o-mini | 1.7324 | 2.0972 | 2.0716 | 2.1556 |
> > | DeepSeek-R1 | 1.2169 | 1.2312 | 1.2451 | 1.2886 |
> >
> > *Experiment B: Fixed Top_p=1.0, Increasing Temperature (Diversity Increases)*
> > | Model | Iteration 1 (t=0.4) | Iteration 2 (t=0.7) | Iteration 3 (t=1.0) | Iteration 4 (t=1.3) |
> > | :--- | :--- | :--- | :--- | :--- |
> > | GPT-4o-mini | 1.7411 | 2.1075 | 2.1236 | 2.1773 |
> > | DeepSeek-R1 | 1.2519 | 1.2919 | 1.2937 | 1.3310 |
> >
> > *Experiment C: Parameters among iterations were identical but values differed, using gpt-4o-mini*
> >
> > | temperature | top_p | Iteration 1 | Iteration 2 | Iteration 3 | Iteration 4 |
> > |-------------|-------|-------------|-------------|-------------|-------------|
> > | 1.0 | 0.4 | 1.7926 | 2.1476 | 2.1019 | 2.1858 |
> > | 1.0 | 0.6 | 1.7587 | 2.0874 | 2.0820 | 2.1041 |
> > | 1.0 | 0.8 | 1.7245 | 2.0471 | 2.0424 | 2.0628 |
> > | 1.0 | 1.0 | 1.6911 | 2.0071 | 1.9829 | 2.0428 |
> > | 0.4 | 1.0 | 1.7757 | 2.1324 | 2.1217 | 2.1654 |
> > | 0.7 | 1.0 | 1.7418 | 2.0472 | 2.0226 | 2.0837 |
> > | 1.0 | 1.0 | 1.6911 | 2.0071 | 1.9829 | 2.0428 |
> > | 1.3 | 1.0 | 1.6579 | 1.9872 | 1.9251 | 2.0027 |
> >
> >
> > The results show that **adjusting sampling parameters does not mitigate bias amplification**. In fact, in settings with lower diversity (higher certainty), bias is often "locked in" and amplified more readily. This indicates the problem is rooted in the nature of model reasoning, not sampling strategies.

---

> > > ### Author Response · Authors · 2025-11-20
> > >
> > > **Question 1:**
> > > A post-hoc normalization step is mentioned for cases of non-compliance. What is the frequency and do they accumulate for specific models?
> > >
> > > **Response to Question 1:**
> > > This step is primarily to ensure mathematical rigor (sum of probabilities equals 1). Statistical analysis shows that the rate of post-hoc normalization across all experiments is **less than 0.5%** (< 5‰). These cases are randomly distributed across models and do not accumulate in specific ones. Therefore, this is a marginal edge case with no statistical impact on our conclusions.
> > >
> > > **Question 2:**
> > > Experimental section stops at diagnosis. While proposed contrarian speculated in Sec. 6, are there any intervention baseline implemented?
> > >
> > > **Response to Question 2:**
> > > While our paper's primary contribution is the **Diagnosis and Quantification** of MAS bias amplification  systemic issue, we implemented a **Fairness Intervention Baseline** to address your concern.
> > > In the chain structure from Figure 4(a), we replaced the L3 Agent with a "Fairness Agent" (prompted to provide critical, balanced viewpoints: Please provide a critical review of the opinions from L1 and L2, presenting challenging or alternative viewpoints, and ensure that your analysis remains balanced, fair, and unbiased.). Results (Absolute Gini):
> > >
> > > | Model | Setting | L1 | L2 | L3 | L4 |
> > > |-------------|-------|-------------|-------------|-------------|-------------|
> > > | **GPT-4o** | No Intervention | 0.0771 | 0.0965 | 0.1054 | 0.1089 |
> > > | | **L3 Fairness Intervention** | 0.0771 | 0.0965 | **0.0215** | 0.0745 |
> > > | **DeepSeek-R1** | No Intervention | 0.2695 | 0.3533 | 0.3657 | 0.3838 |
> > > | | **L3 Fairness Intervention** | 0.2695 | 0.3533 | **0.1127** | 0.2852 |
> > >
> > > The results show:
> > > 1.  Intervention is effective locally; L3's bias drops significantly.
> > > 2.  Crucially, however, **bias is stubborn**. Despite L3 outputting fair reasoning, L4 (considering history from L1/L2) sees bias levels rise again (settling between L2 and L1). This confirms that "bias amplification" is a highly resilient systemic dynamic, further underscoring the importance of our study.

---

### Author Response · Authors · 2025-11-20

We sincerely thank the three reviewers for their detailed reviews and constructive feedback. We are particularly encouraged by the recognition of the value of the **Discrim-Eval-Open benchmark** and the **qualitative findings regarding "neutral triggers"** revealing system fragility.

In this response, we address concerns regarding **bias definitions, amplification mechanisms, metric selection, and model diversity** through extensive supplementary experiments (including ablation studies, broader model combinations, parameter sensitivity analyses, and targeted intervention baselines). All empirical results strongly support our core conclusions. We will integrate all new results into the final revision of the paper.

---

### Meta-Review · Area_Chair_y9Hh · 2025-12-16

**Summary:**

This paper systematically quantifies that in LLM-based multi-agent systems (MAS), even if individual agents are relatively neutral, biases related to gender, age, race, etc., can be amplified through interaction. Using Discrim-Eval-Open (controlled 3-option/probability output) and inequality metrics (such as Gini), the paper verifies layer-wise amplification analysis and robustness against prompt, sampling, and model diversity. In the author response, major concerns were empirically addressed by presenting layer-wise amplification coefficients, conducting qualitative/quantitative follow-up experiments on mechanisms (conformity vs. noise), and adding interventions (such as Fairness Agents).

**Reviewer Concerns:**

- **Validity of the bias definition ("deviation from uniform distribution")** (4dTK): Most concerns were resolved by clarifying the *ceteris paribus* nature of the setup (controlling for health/situational factors, with sensitive attributes being the only difference).
- **Insufficient explanation/validity of metrics (Gini, etc.) and concerns about "LLM-as-a-judge" reliance** (EpdQ): Improved by clarifying that the metrics are deterministic numerical calculations derived from probability distributions rather than a "judge," and by adding intuition, references, and auxiliary metrics.
- **Amplification Mechanism (Conformity vs. Noise)** (4dTK): The conformity hypothesis is further supported by rationale tracking in case studies and neutralization ablation studies (re-amplification occurs when upstream bias is visible).
- **Setting Dependency (Does it disappear with prompts, Temperature/Top-p, or model diversity?)** (U8T3, 4dTK): Robustness is reinforced through sensitive word removal ablations, parameter sensitivity analysis, and follow-up experiments on model combinations.
- **Depth of Mitigation Strategies** (EpdQ, et al.): While local reduction via Fairness Agents is demonstrated, the systemization of practical defenses—including the potential for bias to return downstream—remains a topic for future work.

**Reviewer Scores:**

- **Reviewer 4dTK (Initial: 4):** **No explicit score update (AC prediction).** Major concerns have been largely resolved by clarifying the definition premises (*ceteris paribus*), reporting layer-wise amplification coefficients, additional verification of conformity vs. noise, and follow-up tests on sampling sensitivity. **4 → 6 (High Confidence)**.
- **Reviewer EpdQ (Initial: 4):** **No explicit score update (AC prediction).** The misunderstanding regarding reliance on "LLM-as-a-judge" has been cleared, and explanations for Gini intuition/references/auxiliary metrics have been reinforced. Although mitigation is limited, additional intervention baselines were provided. **4 → 6 (Medium-High Confidence)**.
- **Reviewer U8T3 (Initial: 6):** **No explicit score update (AC prediction).** While the theoretical explanation remains somewhat thin, the robustness of the claims is strengthened by the presented follow-up experiments. **Maintained 6 → 6 (Medium-High Confidence)**.

**Predicted Average:** (6 + 6 + 6) / 3 = **6.0**

---

### Decision · Program_Chairs · 2026-01-26

Accept (Poster)